# Evaluation of Cardiac Autonomic Function in Patients Undergoing Thoracoscopic Sympathetic Chain Clamping for Primary Focal Hyperhidrosis

**DOI:** 10.3390/medsci13030147

**Published:** 2025-08-20

**Authors:** Danilo Ricciardi, Daniele Valente, Paola Liporace, Enrico Davoli, Elisabetta Sposito, Francesco Picarelli, Flavio Angelo Gioia, Vito Calabrese, Gian Paolo Ussia, Francesco Grigioni

**Affiliations:** 1Research Unit of Cardiovascular Science, Department of Medicine and Surgery, Università Campus Bio-Medico di Roma, 00128 Rome, Italy; d.ricciardi@policlinicocampus.it (D.R.); paola.liporace@unicampus.it (P.L.); elisabettasposito5@gmail.com (E.S.); f.picarelli@policlinicocampus.it (F.P.); f.gioia@policlinicocampus.it (F.A.G.); v.calabrese@policlinicocampus.it (V.C.); g.ussia@policlinicocampus.it (G.P.U.); f.grigioni@policlinicocampus.it (F.G.); 2Cardiology Unit, Fondazione Policlinico Universitario Campus Bio-Medico, 00128 Rome, Italy; 3Department of Thoracic Surgery, Fondazione Policlinico Universitario Campus Bio-Medico, 00128 Rome, Italy; e.davoli@policlinicocampus.it

**Keywords:** HRV, PFHH, ECG, heart rate variability, sympathetic tone, thoracoscopic sympathetic chain clamping

## Abstract

**Background/Objectives**. Heart rate variability (HRV) is the variability in the beat-by-beat heart period. Primary focal hyperhidrosis (PFHH) is a disease characterized by excessive sweat production, strongly affecting social life. Several authors define this condition as a dysautonomic disorder, mainly driven by exaggerated sympathetic activity. The aim of the study was to demonstrate a possible cardiac involvement in the disease. Other outcomes were the occurrence of dysautonomic disorders after surgery and its possible correlation with baseline characteristics. **Methods**. This observational, controlled trial enrolled patients with a confirmed clinical diagnosis of severe PFHH candidates to thoracoscopic sympathetic chain clamping. Before and after surgery, ECG was obtained using KardiaMobile 6L (AliveCor^®^, Mountain View, CA, USA) device with a five-minute recording and HRV was analyzed using Kubios HRV Premium (Kubios©) software. **Results**. 111 patients were compared to 222 healthy control subjects. No differences were seen in HRV analysis between the two groups at baseline (time-domain *p* > 0.05, frequency-domain *p* > 0.05, autonomic indexes *p* > 0.05). When comparing autonomic function indexes in patients before and after the surgical procedure, no differences were seen in frequency-domain HRV analysis, but a blunted increase in SNS index (0.2 vs. 1.38, *p* 0.02). No development of systemic dysautonomic disorders nor significant compensatory hyperhidrosis were seen after the surgery. **Conclusions**. This study shows that PFHH is a peripheral autonomic nervous system derangement, rather than central. Sympathetic chain clamping resulted safe and effective in improving patients’ conditions, with no risks of dysautonomic disorders.

## 1. Introduction

The production and secretion of sweat are controlled by the sympathetic arm of the autonomic nervous system (ANS) [1,2]. Therefore, dysfunction in ANS may cause sweating disorders [3]. Primary focal hyperhidrosis (PFHH) is a complex disease, characterized by excessive sweat production from eccrine sweat glands, causing detrimental effects on patient’s psychological health, strongly impacting on social life and activity of daily living [4]. Several reports describe an incidence ranging from 0.13% in Europe to 0.28% in the USA [5]. The pathophysiology of this disorder is still controversial, but several authors define this condition as a dysautonomic disorder, mainly characterized by exaggerated sympathetic activity [6]. Neither cutaneous histopathological changes, nor increase in gland number have been observed in these patients [7]. Endoscopic thoracic sympathectomy is used in severe cases of hyperhidrosis not responding to topical or medical therapies [8].

The clipping of the thoracic sympathetic chain is a safe and reversible surgical technique, with good results and a low complication rate, as demonstrated by Sciucchetti et al. [9]. Since thoracic chain is near the cardiac ganglia, it is correct to assume that the clamping procedure may affect also autonomic cardiac modulation.

Heart Rate Variability (HRV) is the amount by which the time interval between heartbeats varies from beat to beat. It reflects the effects of sympathetic and parasympathetic systems and their interaction with the heart, in particular, their interplay on sinoatrial node (SAN) function modulation: vagal action determines high HRV, instead adrenergic action causes low HRV [10]. Many studies have described HRV measurements, their meaning and applications [11,12,13,14].

There are few studies in literature about HRV in PFHH patients, whose results are controversial and inconsistent, mainly because of poor sample sizes [15,16,17] and even fewer evaluated this association in those who experienced compensatory hyperidrosis after sympathectomy [17,18]. No relevant recent studies were found. The present study aims to investigate the entity of sympathetic and parasympathetic action via HRV measurements in PFHH patients candidate for thoracoscopic sympathetic chain clamping versus healthy control subjects, to better understand PFHH pathophysiology. The safety and efficacy of surgical procedure and the occurrence of dysautonomic disorders after surgery were also evaluated.

## 2. Materials and Methods

### 2.1. Study Design

The study was a single-center (at Fondazione Policlinico Universitario Campus Bio-Medico), prospective, non-randomized, case-control trial that analyzed cardiac autonomic activity in patients with PFHH eligible for thoracic sympathetic chain clamping (case group) versus healthy individuals (control group). Data as clinical history, physical examination, ECG recordings–further described in the next sections-were obtained from PFHH patients during pre-surgical screenings, while data from controls were collected from people who freely volunteered to participate in the trial. This study was approved by the Ethics Committee of Fondazione Policlinico Universitario Campus Bio-Medico, approval number [2022.231, 31 May 2023]. All procedures were under the ethical standards of the 1964 Declaration of Helsinki and its later amendments.

### 2.2. Patients

Consecutive patients with confirmed clinical diagnosis of severe facial and/or axillary PFHH eligible for thoracoscopic sympathetic chain clamping in sinus rhythm s, aged 18 to 50 years, with the ability to understand and provide written consent were enrolled. Only ECG recordings acquired with the KardiaMobile^®^ 6L (AliveCor Inc., Mountain View, CA, USA) device that were free from noise artifacts and clearly interpretable were included in the analysis. Exclusion criteria were: patients with feet or urogenital PFHH, patients under medications that could modify heart rate or heart variability, atrial fibrillation, flutter, sinoatrial dysfunction or more than 20% of premature ventricular complexes during the recording, cardiovascular diseases, other dysautonomic syndromes. A group of age-matched healthy volunteers without any history of cardiovascular, neurological, or autonomic disorders was recruited as the control group. The same ECG acquisition and quality criteria were applied to the control group. All patients provided written informed consent.

### 2.3. Procedures

#### 2.3.1. ECG and HRV Recording

Patients underwent a cardiologic evaluation and ECG monitoring prior to surgery. Resting ECGs were obtained using KardiaMobile^®^ 6L (AliveCor Inc.) device with a five-minute recording in a quiet room, to minimize external stimuli that could influence autonomic tone and affect HRV results. The same acquisition protocol and quality criteria were applied to the control group. Post-operative ECGs were obtained the day after surgery, following the same procedure and under the same environmental conditions. HRV was analyzed using Kubios© HRV Premium (3.5.0 version, May 2021, Varsitie 22, 70150 Kuopio, Finland). Preprocessing of HRV data was performed using automatic “noise detection”, “beat correction” and “trend removal” programs. Subsequently, each recording was manually reviewed by an operator to ensure that no residual errors were present. Time-domain and frequency-domain analyses were performed. Mean RR interval, standard deviation of RR intervals (SDNN), root mean square of successive RR interval differences (RMSSD) and relative number of successive intervals differing by more than 50 msec (pNN50) were derived from time-domain analysis. Very low frequency (VLF), low frequency (LF) and high frequency (HF) peaks, their absolute powers and LF/HF ratio were derived from frequency-domain analysis. PSN and SNS indexes were automatically derived by the software using integrated parameters. For PSN, mean RR, RMSSD, and Poincaré SD1 were used; for SNS, mean HR, Baevsky’s Stress Index, and Poincaré SD2. For a comprehensive description of the HRV parameters, their calculation, and physiological relevance, readers are referred to the cited literature and to the Summary Table 1 [11,12,13,14]. ECG features were also sampled, including PR interval length, QRS duration, QT interval, and the corrected QT interval (QTc), computed according to Bazett’s formula.

##### Thoracoscopic Sympathetic Chain Clamping

Each patient underwent surgical procedure of: “Selective Clamping of the Thoracic Nerve in Video-Thoracoscopy”. This procedure, an update of the widely known ETS (Endoscopic Thoracic Sympathectomy) practically consists in narrowing the Nerve by placing titanium clips over and below a specific ganglium, along the white communicating rami, carefully sparing the ganglia and the efferent rami to the cardiac plexus. The specific ganglium is selected according to the body area mainly affected by PFHH: generally, T2 and T3 in case of cranio-facial Hyperhidrosis, T3 for hands and T4 for underarms [19].

The Thoracoscopy is performed under general anesthesia, although is a minimally invasive technique, with the use of two tiny (5 mm in diameter) atraumatic trocars introduced through the fourth and 6ixth intercostal spaces on mid axillary line. After obtaining a pneumothorax with the introduction of CO_2_ gas, the sympathetic chain is clamped through a 3 mm incision of the posterior parietal pleura. All patients are discharged on the following day, after a chest x-ray, in order to exclude pneumothorax and to verify the correct clip placement.

##### Satisfaction Questionnaire

A questionnaire was drawn up, to assess patients’ outcomes and satisfaction, the occurrence of compensatory hyperhidrosis and the development of dysautonomic disorders after the procedure. For dysautonomic evaluation, we used COMPASS-31 for assessing orthostatic intolerance, gastroparesis, constipation, diarrhea, bladder disorders, syncope, palpitations and ocular dysfunction [20].

The questionnaire was administered to all PFHH patients six months after surgery, as an online survey. A copy of the questionnaire is available in Appendix A.

### 2.4. Outcomes

The primary outcome of the study was to evaluate whether PFHH patients exhibit alterations in ANS function compared to healthy controls. We assessed this through a comparative analysis of all HRV parameters previously described and ECG features between the two groups. Secondary outcomes were the occurrence of potential changes in ANS function following thoracic sympathetic chain clamping and the incidence of postoperative dysautonomic disorders. Additionaly, our study explored whether these changes were associated with baseline characteristics.

### 2.5. Statistical Analysis

Normally distributed data were expressed as mean (±SD). Collected data were compared using the two-sided Student’s t test with a confidence interval of 95% and a two-sided 0.05 type I error. A priori power analysis was conducted to determine the minimum sample size required to detect a statistically significant difference in HRV between PFHH patients and healthy controls. Assuming a mean difference of =0.8 in SNS and PNS between the two groups, α = 0.05, and power (1 − β) = 0.80, a total of 26 subjects per group were required. The final sample size exceeded this threshold, ensuring adequate power to detect clinically relevant differences.

Fisher’s exact test was used to compare differences in categorical variables, with a *p* value of 0.05. The direct dependence of dysautonomic disorders was studied using Spearman’s correlation. All statistical analysis was performed with the use of R (version 4.3.3) and STATA18 MP (STATACORP LLC©, version 18.0, StataCorp, 4905 Lakeway Drive, College Station, TX, USA).

## 3. Results

### 3.1. Baseline Characteristics

Between March 2023 and December 2023, 333 individuals were enrolled, 111 PFHH patients (case group) and 222 controls. Baseline characteristics are shown in Table 2 and Table 3. Mean age was 30.4 (±10.35) in PFHH group and 31.97 (±11.87) in control group. On baseline ECG, PFHH group showed longer QRS if compared to control group (94.75 ± 10.43 vs. 85.42 ± 9.29, *p* > 0.05). Corrected QT interval was slightly shorter in PFHH group (398.71 ± 17.98 vs. 403.77 ± 28.99, *p* = 0.05). There were no differences in PR interval between the two groups.

No differences were seen in time-domain HRV analysis and autonomic function indexes between the two groups, as shown in Table 3. Frequency-domain HRV analysis showed lower VLF (82.8 ± 76.37 vs. 133.12 ± 123.31, *p* = 0.006) and LF power (1147.53 ± 1230.25 vs. 1457.36 ± 1433.63, *p* = 0.04) in PFHH group than controls. LF/HF ratio was 2.54 ± 2.74 in PFHH group and 3.11 ± 2.89 in controls with no significant differences (*p* = 0.08).

Then we compared HRV analysis before and after surgery. The results can be seen in Table 4. If compared using autonomic function indexes, the pre-operative PFHH group showed higher PNS (0.24 ± 1.21 vs. −0.7 ± 1.11, *p* = 0.01) and lower SNS (0.2 ± 1.08 vs. 1.38 ± 2.09, *p* = 0.02) than the post-operative PFHH group. No differences were seen in frequency-domain HRV analysis.

### 3.2. Compensatory Hyperhidrosis (CH)

Six months after surgery, a survey was administered to all 111 patients for the evaluation of compensatory hyperhidrosis. According to the questionnaire results, 28 patients developed compensatory hyperhidrosis, but only 7 (6.3%) of them reported an absolute absence of improvement or even worsening after surgery. The patients were requested to assess the discomfort experienced depending on the different sites of sweating. Back CH appeared to be the main site of discomfort.

Dysautonomic changes after surgery were also assessed through the questionnaire and no differences were observed (Table 5).

### 3.3. Other Results

A subgroup HRV analysis was performed between those who experienced compensatory hyperhidrosis and (28 patients) and those who did not. No significant differences resulted, apart from PR segment length, which seemed to be shorter in those who did not develop compensatory hyperhidrosis (139.46 ± 11.49 vs. −154.25 ± 22.14, *p* = 0.029). All results are listed in Table 6.

## 4. Discussion

PFHH is a condition characterized by excessive sweating, causing discomfort and impacting the activity of daily living, bringing upon severe social and psychological constraints [21]. PFHH is frequently generalized, but often it could be selectively localized on facial areas or on the back [22]. The etiology of PFHH is still object of debate among the Experts, but there is a collective agreement that it could be an expression of autonomic derangement, in particular, some Authors attribute this disorder to a sympathetic neurogenic hyperactivation of eccrine glands [6]. Therefore, it is reasonable to consider sympathetic chain clamping the most effective and radical treatment.

HRV is a physiological phenomenon that reflects ANS action on the sinoatrial node. Sympathetic nervous system (SNS) increases heart rate and, in doing so, reduces HRV; while parasympathetic nervous system (PNS) slows heart rate and increases HRV [12]. The analysis is based on three methods: time-domain, frequency-domain and PNS/SNS indexes [23,24,25].

As PFHH is thought to be mainly driven by SNS hyperactivity, it is reasonable to think that even heart rate could be affected and, therefore, lower HRV levels and high SNS index are expected. A few studies have been conducted on this subject and the results are controversial. Birner P et al. compared 63 PFHH patients with 28 healthy subjects using frequency domain power spectral analysis, finding no concrete evidence of cardiac sympathetic dysfunction, but a marked parasympathetic dysfunction in hyperhidrosis patients was observed [15]. A study by Kaya D et al. investigated cardiac autonomic function during controlled respiration in 12 subjects with PFHH using 20 healthy subjects as a control group. They found no differences between both groups at baseline, whereas an increase in heart rate during controlled respiration was not seen in case group as expected, showing that the parasympathetic system seems to be involved in the pathogenesis of this disorder [26]. No differences were found even in another study by Niwa A et al., who selected 34 PFHH patients and 34 controls for HRV analysis [16].

Our study enrolled the largest sample size in literature. We obtained ECG recordings of 333 subjects (111 cases and 222 controls) and analyzed them in terms of time and frequency domain methods and PNS/SNS indexes and we found no significant differences in baseline characteristics.

One important thing to notice is the LF/HF ratio. It defines the relationship between low frequencies oscillation (LF), mainly influenced by SNS, and high-frequency bands that are known to be regulated by PNS [27]. Its prognostic value is controversial and still not validated. Moon DH and colleagues conducted a study to evaluate ANS in 53 PFHH patients undergoing endoscopic thoracic sympathectomy. Their aim was to evaluate whether this parameter could have a prognostic value in predicting compensatory hyperhidrosis after surgery. They found that if LF/HF ratio was less than 0.66, there was a parasympathetic predominance; if more than 2.60, there was a sympathetic predominance, whereas between values were considered as a balanced effect of ANS components [28]. However, these cut-off values were based on a sample size way too small to be generalized to all population. In fact, in our study we found contrasting results about this topic: the LF/HF ratio was 2.54 ± 2.74 in case group and 3.11 ± 2.89 in controls. Our results did not support the evidence portraited by Moon DH and colleagues. Using these cut-offs, our population seem to be more sympathetic, regardless of being in PFHH or in control group. Anyway, when we compared HRV analysis before and after surgery, we found interesting results. Before surgery, patients seemed to have a balanced response between PNS and SNS, while after surgery they tended to be more sympathetic driven. In fact, post-surgery SNS index was higher (1.38 vs. 0.2, *p* = 0.02). This evidence is supported by other results suggesting a marked sympathetic activity in this population, such as MeanHR, MeanRR, SDNN and RMSSD, as shown in Table 4. Based on these results, our hypothesis is that PFHH patients have a pronounced parasympathetic activity as a compensatory mechanism to counteract sympathetic effects, leading to a pseudo-normal and balanced ANS activity. When thoracic sympathetic chain gets clamped, somehow, SNS exaggerates its triggers and overcomes parasympathetic activity. To date, no evidence exists to either support or contradict our hypothesis, which thus remains merely speculative. Other studies are necessary to confirm.

CH is a serious and relatively common side effect after surgery, that often results in a reduced quality of life and post-intervention satisfaction [29,30]. Several attempts were made to avoid or, at least, reduce the occurrence of CH, but it remains around 10 to 50%, with a lower incidence when reversible surgery with clipping rather than sympathectomy is performed [31]. We designed a survey with questions about occurrence of CH, impact on quality of life, dysautonomic disturbances and surgery-satisfaction level [32]. Out of 111 patients, around 28 patients reported CH, but only 7 (6.3%) of them reported severe CH with absolute absence of improvement or even worsening of quality of life after surgery. When comparing these 28 patients with 14 patients who reported considerable improvement in quality of life after surgery, no statistically significant differences in HRV parameters were found.

Three studies reported that palmar, thoracic, back, craniofacial and axillary CH were the most common types that harshly impact quality of life [33,34,35]. In our experience, we found back CH is the most frequent type, followed by abdominal, axillary and thoracic types. Whilst patients affected by CH could have not experienced sweating in some body areas, back hyperhidrosis was always referred and was associated with worst outcomes.

Although thoracic sympathetic chain clamping is a well-established treatment to manage severe PFHH, it carries several consequences. Besides surgical site complications, bleeding and pneumothorax, this type of surgery permanently cuts off the sympathetic chain, usually somewhere between T2 and T4 ganglia, potentially causing systemic autonomic function imbalances, although quite rare [35,36,37]. Apart from CH or dry hands, several studies reported, as major consequence of this procedure, a change in cardiac autonomic regulation, causing a marked shift of sympathovagal balance toward parasympathetic tone [38]. No autonomic imbalance in heart function was observed in our population.

Lastly, we included in our questionnaire some questions about the occurrence of dysautonomic disorders after surgery. A few gastrointestinal, urinary and thermoregulatory disorders were reported, but none seemed to be related to the procedure. 51 patients reported pronounced sweating right after the ingestion of spicy food or hot beverages (Frey’s syndrome) but did not reach statistical significance (*p* = 0.0507). Therefore, in our experience, clamping procedure resulted as safe as endoscopic thoracic sympathectomy and it was associated with a better quality of life.

There are several limitations to this study. First, the degree of CH was measured subjectively, through a non-supervised post-surgery satisfaction questionnaire. However, up to date, there is no objective method of quantifying the degree or severity of CH. Second, given the HRV being variable across time, the results presented may have small measurement errors. To avoid this, we attempted to perform ECG recording under the same conditions for all candidates, trying to exclude possible external conditions that may influence HRV results. However, we acknowledge that certain confounding factors—such as caffeine intake, physical activity, and circadian influences, all of which are known to affect HRV—were not strictly controlled in this study. These aspects represent additional limitations and should be addressed in future research to strengthen the reliability of HRV measurements. Furthermore, our study was a single-center observational study with short-time follow-up. A future study including more centers and longer follow-ups may be interesting.

## 5. Conclusions

Our population suggests that PFHH is a complex disorder, whose pathophysiology seems to be determined by peripheral autonomic nervous system derangement rather than a systemic autonomic disorder. Sympathetic chain clamping resulted safe and effective in improving patients’ conditions, with no risks of dysautonomic disorders. CH is a relatively common consequence after surgery, but only in a few cases are severe and responsible for worsening quality of life.

## Figures and Tables

**Table 1 medsci-13-00147-t001:** Overview of the main HRV parameters mentioned in the text, grouped by domain. Time-, frequency-, and nonlinear-domain metrics are listed along with their definitions and physiological relevance.

Domain	Parameter	Units	Description
Time-domain	Mean RR	ms	Average interval between consecutive heartbeats (RR intervals)
	SDNN	ms	Standard deviation of RR intervals–reflects overall HRV
	RMSSD	ms	Root mean square of successive RR interval differences–parasympathetic marker
	pNN50	%	Percentage of successive RR intervals differing by >50 ms
	Mean HR	bpm	Average heart rate over the recording period
Frequency-domain	VLF (Very Low Frequency)	ms^2^ or nu (normalized units)	Power in the 0.003–0.04 Hz band–physiological interpretation still debated
	LF (Low Frequency)	ms² or nu	Power in the 0.04–0.15 Hz band–reflects both sympathetic and parasympathetic activity
	HF (High Frequency)	ms^2^ or nu	Power in the 0.15–0.4 Hz band–represents parasympathetic (vagal) activity
	LF/HF ratio	-	Ratio of LF to HF power–used as an index of sympathovagal balance
Nonlinear	SD1 (Poincaré)	ms	Short-term variability–vagal modulation
	SD2 (Poincaré)	ms	Long-term variability–associated with sympathetic activity
	Baevsky’s Stress Index	-	Index derived from histogram of RR intervals–reflects sympathetic dominance
	PSN Index	-	Parasympathetic nervous system index–derived from Mean RR, RMSSD, and SD1
	SNS Index	-	Sympathetic nervous system index–derived from Mean HR, Baevsky’s Index, and SD2

**Table 2 medsci-13-00147-t002:** Baseline clinical and ECG characteristics.

	PFHH	Control	*p*-Value
N	111	222	
Age (years)	30.4 ± 10.35	31.97 ± 11.87	0.17
BMI (Kg/m^2^)	23.24 ± 3.27	22.82 ± 3.71	0.30
MEAN HR (bpm)	78.25 ± 13.41	77.04 ± 12.37	0.43
MIN HR (bpm)	68.78 ± 11.8	67.50 ± 11.01	0.35
MAX HR (bpm)	90.91 ± 15.94	90.45 ± 14.26	0.8
PR (ms)	151.36 ± 19.01	155.74 ± 20.68	0.06
QRS (ms)	94.75 ± 10.43	85.42 ± 9.29	>0.01
QTc (ms)	398.71 ± 17.98	403.77 ± 28.99	0.05

**Table 3 medsci-13-00147-t003:** Baseline characteristics.

	PFHH	Control	*p*-Value
Time-Domain HRV Analysis Methods			
MEAN RR (ms)	787.6 ± 129.03	799.1 ± 130.85	0.45
SDNN (ms)	44.9 ± 23.27	46.9 ± 23.85	0.46
RMSSD (ms)	43.8 ± 30.55	43.54 ± 29.91	0.93
NN50 (beats)	43 ± 44.1	52.9 ± 47.09	0.06
pNN50 (%)	20.54 ± 20.03	20.65 ± 18.91	0.95
Autonomic function indexes			
PNS	−0.55 ± 1.34	−0.52 ± 1.33	0.84
SNS	1.33 ± 1.86	1.18 ± 1.71	0.48
Frequency-domain HRV analysis methods			
Peak VLF (Hz)	0.034 ± 0.005	0.035 ± 0.016	0.57
Peak LF (Hz)	0.09 ± 0.03	0.086 ± 0.028	0.31
Peak HF (Hz)	0.23 ± 0.072	0.22 ± 0.077	0.49
VLF Power (ms^2^)	82.8 ± 76.37	133.12 ± 123.31	0.006
LF Power (ms^2^)	1147.5 ± 1230.25	1457.3 ± 1433.63	0.04
HF Power (ms^2^)	1099 ± 1804.57	986 ± 1238.84	0.57
Total Power (ms^2^)	2330 ± 28.36	2552 ± 2690.51	0.5
LF/HF	2.54 ± 2.74	3.11 ± 2.89	0.08

**Table 4 medsci-13-00147-t004:** Before surgery HRV analysis and baseline characteristics compared with after surgery results.

	Before Surgery	After Surgery	*p*-Value
N	111	111	
MEAN HR (bpm)	70.5 ± 10.19	78.1 ± 12.39	0.04
MIN HR (bpm)	62.3 ± 8.9	68.3 ± 11.42	0.06
MAX HR (bpm)	84 ± 14.21	92.5 ± 14.73	0.06
PR (ms)	156 ± 20.35	152 ± 18.82	0.53
QRS (ms)	93.6 ± 10.62	93 ± 10.67	0.85
QT (ms)	390 ± 27.5	384 ± 26.85	0.46
QTc (ms)	403.7 ± 16.36	399.9 ± 16.69	0.43
Time-domain HRV analysis methods			
MEAN RR (ms)	869.3 ± 138.34	784.7 ± 111.87	0.03
SDNN (ms)	53.32 ± 18.57	40.68 ± 17.55	0.03
RMSSD (ms)	57.64 ± 28.93	38.77 ± 22.85	0.02
NN50 (beats)	69.05 ± 20.92	48.58 ± 44.46	0.17
pNN50 (%)	29.22 ± 20.92	17.51 ± 17.92	0.06
Autonomic function indexes			
PNS	0.24 ± 1.21	−0.7 ± 1.11	0.01
SNS	0.2 ± 1.08	1.38 ± 2.09	0.02
Frequency-domain HRV analysis methods			
Peak VLF (Hz)	0.034 ± 0,006	0.033 ± 0.004	0.5
Peak LF (Hz)	0.079 ± 0.03	0.09 ± 0.02	0.12
Peak HF (Hz)	0.23 ± 0.079	0.23 ± 0.07	0.76
VLF Power (ms^2^)	144.3 ± 178.83	110.8 ± 105.86	0.48
LF Power (ms^2^)	1248 ± 1041.92	1025 ± 888.45	0.45
HF Power (ms^2^)	1260 ± 1498.43	755.8 ± 840.17	0.17
Total Power (ms^2^)	2654.7 ± 1972.57	1892 ± 1522.73	0.16
LF/HF	2.59 ± 2.92	2.71 ± 2.88	0.9

**Table 5 medsci-13-00147-t005:** Dysautonomic disorder development after surgery.

Dysautonomic Manifestation		*p*-Value	Dysautonomic Manifestation		*p*-Value
Thermoregulation	No changes (34)	0.454	Gastrointestinal Disorders	No changes (68)	0.301
	Feeling warmer (32)			Functional dyspepsia (2)	
	Feeling cooler (20)			Meteorism (9)	
Food sweating	No (51)	0.058		Constipation (2)	
	Yes (35)			Diarrhea (2)	
Rash	No (74)	0.482		Sickness and vomiting (2)	
	Yes (12)			*No answer (1)*	
GERD	No changes (75)	0.852	Urinary tract Disorders	No changes (82)	0.103
	Improved (4)			Urinary retention (2)	
	Worsened (3)			Incontinence (1)	
	Developed (4)			*No answer (1)*	
Stamina	No changes (72)	0.10	Eye Disorders	No changes (77)	0.766
	Improved (3)			Photophobia (5)	
	Worsened (11)			Accommodative disorder (3)	
Breathing	No changes (79)	0.418		*No answer (1)*	
	Improved (2)		Palpitation	No (69)	0.283
	Worsened (5)			Yes (16)	
Focus	No changes (71)	0.619		*No answer (1)*	
	Improved (12)		Syncope	No (84)	0.683
				Yes (1)	
	Worsened (3)			*No answer (1)*	

**Table 6 medsci-13-00147-t006:** Baseline pre-surgery HRV characterists comparison between those who experienced CH and those who did not.

	NO CH	CH	*p*-Value
N	83	28	
Age (years)	37.83 ± 20.41	32.87 ± 10.48	0.30
BMI (Kg/m^2^)	23.50 ± 4.01	23.34 ± 3.49	0.89
MEAN HR (bpm)	82.7 ± 14.31	77.6 ± 14.45	0.29
MIN HR (bpm)	72.6 ± 13.82	68.8 ± 13.49	0.40
MAX HR (bpm)	97.1 ± 17.62	90.4 ± 16.01	0.23
PR (ms)	139.46 ± 11.49	154.25 ± 22.14	0.029
QRS (ms)	95 ± 12.51	97.71 ± 16.21	0.59
QT (ms)	376.4 ± 20.26	364.5 ± 81.8	0.60
QTc (ms)	399.5 ± 17.14	396.9 ± 22.46	0.71
Time-domain HRV analysis methods			
MEAN RR (ms)	743.7 ± 123.18	793.7 ± 119.94	0.22
SDNN (ms)	39.1 ± 16.66	42.2 ± 21.36	0.65
RMSSD (ms)	36.3 ± 18.08	39.2 ± 22.07	0.67
NN50 (beats)	47.2 ± 46.08	44.4 ± 41.35	0.84
pNN50 (%)	16.3 ± 13.81	18.6 ± 16.30	0.66
Autonomic function indexes			
PNS	−0.94 ± 1	−0.66 ± 1.12	0.45
SNS	1.92 ± 2.62	1.50 ± 2.20	0.59
Frequency-domain HRV analysis methods			
Peak VLF (Hz)	0.036 ± 0.004	0.034 ± 0.004	0.17
Peak LF (Hz)	0.088 ± 0.03	0.095 ± 0.02	0.48
Peak HF (Hz)	0.22 ± 0.066	0.21 ± 0.065	0.61
VLF Power (ms^2^)	82.7 ± 69.18	78.17 ± 71.23	0.85
LF Power (ms^2^)	854 ± 600.15	1207 ± 1527.9	0.42
HF Power (ms^2^)	483 ± 394.78	900 ± 955.68	0.14
Total Power (ms^2^)	1418.8 ± 938.6	2188.7 ± 2348.87	0.26
LF/HF	3.08 ± 2.79	2.65 ± 2.92	0.66

## Data Availability

The original contributions presented in this study are included in the article/Appendix A. Further inquiries can be directed to the corresponding author(s).

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
