# Peer review of "Evaluation of Cardiac Autonomic Function in Patients Undergoing Thoracoscopic Sympathetic Chain Clamping for Primary Focal Hyperhidrosis"

_medsci, 2025, doi:10.3390/medsci13030147_

Round 1

Reviewer 1 Report

Comments and Suggestions for Authors

This prospective case-control study explores the cardiac autonomic profile in patients with primary focal hyperhidrosis (PFHH) using HRV analysis - an area with limited prior investigation. By comparing 111 PFHH patients to 222 healthy controls and evaluating pre- and post-operative HRV following thoracoscopic sympathetic chain clamping, the authors suggest that PFHH may involve peripheral rather than systemic autonomic dysfunction. Despite minimal HRV differences, the study provides original clinical evidence supporting the safety and autonomic neutrality of the surgical intervention. Its publication may contribute valuable data to better understand PFHH pathophysiology and support clinical decisions regarding surgical treatment.

A detailed list of revision comments requiring careful consideration is provided below.

Revision remarks:

  1. Abstract: Must be rewritten with the following headings: Background/Objectives, Methods, Results, and Conclusions.
  2. Abstract: The phrases “Before surgery, ECG was obtained” and “in patients before and after the surgical procedure” are unclear, as the Methods section does not explicitly define or describe any post-operative ECG acquisition. This inconsistency creates confusion about the timing and scope of the ECG recordings. The methodology must clearly specify whether ECGs were recorded post-surgery and, if so, under what conditions and at what time points.
  3. Abstract: The statement “No differences were seen in HRV analysis” is vague and lacks clarity. It is not specified what type of heart rate variability (HRV) analysis was performed - whether it includes both time-domain and frequency-domain measures of RR intervals specific for the sympathetic and parasympathetic activity, or whether additional nonlinear methods such as Poincaré plot analysis were considered. Furthermore, the abstract omits key numerical results relevant to inter-group comparisons. I highly recommend adding quantitative results to improve reader’s ability to assess the depth and rigor of the study’s analytical approach.
  4. Abstract: The statement “Sympathetic chain clamping resulted safe and effective in improving patients’ conditions” is not supported by the results presented in the abstract. The findings indicate only non-significant differences in HRV before and after the surgical procedure, without providing any specific evidence of clinical improvement. Therefore, the conclusion appears overstated and should be revised to accurately reflect the reported data.
  5. Introduction: The investigation of the problem seems too old as references are limited up to 2023, most of them published before 2020. I strongly recommend update of the research analysis of published literature in order to show the state-of-the-art developments in recent years 2024, 2025, considering that the great interest to the automated HRV analysis in the era of AI.
  6. Introduction: “The production and secretion of sweat are controlled by the parasympathetic arm of the autonomic nervous system (ANS).” -> Please revise the statement as there are evidences that the production and secretion of sweat are primarily controlled by the sympathetic branch of the autonomic nervous system, specifically via cholinergic fibers innervating the eccrine sweat glands. Although acetylcholine is typically associated with the parasympathetic system, in the case of sweating it is released by sympathetic postganglionic neurons, making this a notable exception.
  7. Ln 60-64: The aims of the study do not consider observation of the effect of the thoracoscopic sympathetic chain clamping as there are evidences for that in the Abstract. Please, revise the aims to represent the different focuses of the study. One option is to distinguish between primary and secondary objectives. Additionally, the aims should highlight the expected clinical benefits or implications of the findings to better justify the relevance and significance of the research.
  8. Ln 75: “high quality ECG signals” -> Please clarify what is meant by this inclusion criterion. The quality of an ECG primarily reflects the acquisition conditions rather than the patient’s clinical status. Since high-quality ECGs can generally be obtained at any time, this criterion requires further explanation. Additionally, the specifics of ECG acquisition—such as the equipment used and recording conditions—should be described before applying this criterion.
  9. “2.1. Study Design” -> Provide information about the center involved in this single-center trial.
  10. Section “2.1. Study design”: provide information about Ethical approval of this study (Institution, ID and date) and its registration in an official register.
  11. Section “2.2. Patients”: The enrollment process and testing procedures for healthy controls are not described. Since this group serves as the reference, detailed information about their selection and assessment should be clearly provided.
  12. Section “2.3.1. ECG and HRV Recording” -> Please provide detailed information about all devices and software used, including their version numbers and the country of origin of the manufacturers.
  13. Ln 84-85: “KardiaMobile® 6L (AliveCor Inc.) device with a five minute recording in a quiet room.” -> Please explain why recording in a quiet room is important and whether it relates specifically to the transmission of ECG data as audio frames. Otherwise, the absence of ambient noise is not a standard requirement for ECG acquisition, which typically emphasizes a resting state rather than environmental sound conditions.
  14. Ln 87-88: “Each recording was double-checked singularly.” -> Please, explain the statement.
  15. Ln 89: All HRV features must be explained with their full names instead of directly using abbreviations.
  16. Ln 91-93: “Parasympathetic nervous system (PSN) and Sympathetic nervous system (SNS) indexes were obtained from integrated computations.” -> The statement requires further clarification. Please provide more detailed information on how these indexes were computed, as they are not standard measures. Including references, formulas and computational methods would help readers better understand the evaluation process.
  17. In Tables 2, 3, 5 many HRV features are reported, however they are NOT defined in Methods. This lacks interpretation of these features in results. Typically, all HRV features should be clearly defined in Methods, ideally accompanied by an example of the HRV analysis process—such as software used, illustrative plots (Time-domain and frequency-domain), and details of spectral computations - to clarify how these features are derived. Currently, the HRV analysis appears as a black box, making the reported features difficult to interpret.
  18. Ln 93: “ECG features were also sampled.” -> The statement requires further clarification, including the computed ECG features under interest.
  19. Ln 95-96: “Selective Clamping of the Thoracic Nerve in Video-Thoracoscopy” -> Provide reference(s) that define the standard protocols or guidelines for this procure to support its validity.
  20. Section “2.4. Outcomes” -> This section should be expanded to include clear and precise definitions of the primary and secondary outcomes. The term “aim” is inappropriate here, as the study aims are already defined elsewhere, so avoid redundant use. It is also unclear why only the “ANS evaluation of PFHH patients” is mentioned, given that healthy controls serve as a reference group. Please clearly define the groups involved in the outcome analyses, including comparisons between PFHH patients and healthy controls, as well as pre- versus post-operative PFHH patients, specifying the features under investigation for each groups.
  21. Section “2.5 Statistical Analysis” -> Provide power analysis to plan the number of required patients to validate the primary hypothesis.
  22. Ln 125-126: “For a small group of patients, pre-second intervention HRV data were available, and so they were compared with pre-first intervention ones. “ -> This statement is confusing as pre-first and pre-second interventions must be defined before in methods. Clarification is strongly required.
  23. Results: Please consider including a CONSORT-style flow diagram to clearly illustrate the progression of patients from enrollment through to final analysis in groups (do not forget pre-first and pre-second interventions), taking into account all inclusion and exclusion criteria. This would enhance transparency regarding patient selection and clarify how the final study population was determined.
  24. Ln 137: The term “PFHH patients (case group)” is unclear. It is preferable to use a consistent and meaningful term such as “PFHH group” throughout the text. Additionally, specify “pre-operative PFHH group” when appropriate to clearly distinguish it from the “post-operative PFHH group” later in the manuscript.
  25. Results: Please include the sample size (number of participants) for each group in all tables, as this information is essential for proper interpretation of the results.
  26. Table 2, Table 3, Table 5 and corresponding text: Limit all numerical values to three or four significant digits - for example, round 1098.86 to 1099, or 47.09 to 47.1. This will enhance readability and eliminate unnecessary precision that does not add meaningful information.
  27. The results in Table 2 are very interesting. The data confirm that patients with PFHH show reduced sympathetic modulation (as evidenced by significantly lower LF and VLF power), while parasympathetic activity remains unchanged. There is no evidence of parasympathetic dominance, but rather of reduced autonomic response overall, especially on the sympathetic side. These results are consistent with the idea of ANS dysregulation but not of overactive resting sympathetic tone - contrary to what might be assumed based on the pathophysiology of PFHH. These data do not support the hypothesis of resting sympathetic hyperactivity in PFHH. The observed autonomic imbalance may reflect altered central regulation in PFHH, although localized sympathetic overactivity (e.g., to sweat glands) may still be present but not detectable through global HRV metrics.
  28. The results in Table 3 are also very interesting. The post-operative changes show that Heart rate ↑, Time-domain HRV ↓, PNS index ↓, SNS index ↑, and moderate (but non-significant) HF power ↓. These findings point to a significant shift toward sympathetic dominance and vagal withdrawal, likely reflecting the autonomic consequences of thoracoscopic sympathetic chain clamping. Importantly, while time-domain and autonomic indexes show strong and significant effects, frequency-domain measures do not reach significance, possibly due to limited sample size or measurement variability.
  29. Table 5, caption -> Not informative.
  30. Results: As there is no restriction on the number of figures or tables, I recommend including graphical representations to enhance the visualization of key findings from the most important tables. This would improve data interpretation and make group comparisons clearer for the reader.
  31. Section “Author Contributions” -> Must be rewritten according to the journal’s guidelines, described here: https://www.mdpi.com/journal/medsci/instructions, copy of the text: “For research articles with several authors, a short paragraph specifying their individual contributions must be provided. The following statements should be used "Conceptualization, X.X. and Y.Y.; Methodology, X.X.; Software, X.X.; Validation, X.X., Y.Y. and Z.Z.; Formal Analysis, X.X.; Investigation, X.X.; Resources, X.X.; Data Curation, X.X.; Writing – Original Draft Preparation, X.X.; Writing – Review & Editing, X.X.; Visualization, X.X.; Supervision, X.X.; Project Administration, X.X.; Funding Acquisition, Y.Y.””

Author Response

We are grateful to the reviewer for his/her thoughtful and detailed suggestions, which have significantly contributed to enhancing the manuscript.

Comment 1: Abstract: Must be rewritten with the following headings: Background/Objectives, Methods, Results, and Conclusions.

Response 1: Thank you for pointing this out. I rectified the abstract.

Comment 2: Abstract: The phrases “Before surgery, ECG was obtained” and “in patients before and after the surgical procedure” are unclear, as the Methods section does not explicitly define or describe any post-operative ECG acquisition. This inconsistency creates confusion about the timing and scope of the ECG recordings. The methodology must clearly specify whether ECGs were recorded post-surgery and, if so, under what conditions and at what time points.

Response 2: Thank you for your observation. You are absolutely right — we had missed this detail in the original manuscript. We have now corrected the issue in the revised version of the text, as you can see on page 2, lines 85-86 and in the Abstract

Comment 3: Abstract: The statement “No differences were seen in HRV analysis” is vague and lacks clarity. It is not specified what type of heart rate variability (HRV) analysis was performed - whether it includes both time-domain and frequency-domain measures of RR intervals specific for the sympathetic and parasympathetic activity, or whether additional nonlinear methods such as Poincaré plot analysis were considered. Furthermore, the abstract omits key numerical results relevant to inter-group comparisons. I highly recommend adding quantitative results to improve reader’s ability to assess the depth and rigor of the study’s analytical approach.

Response 3: We thank the reviewer for the comment. The information in question is already included in the manuscript (page 3, lines 89–96).  However, to improve clarity, we have slightly revised the Abstract to make this point more explicit.

Comment 4: Abstract: The statement “Sympathetic chain clamping resulted safe and effective in improving patients’ conditions” is not supported by the results presented in the abstract. The findings indicate only non-significant differences in HRV before and after the surgical procedure, without providing any specific evidence of clinical improvement. Therefore, the conclusion appears overstated and should be revised to accurately reflect the reported data.

Response 4: We thank the reviewer for this important observation. We respectfully clarify that the statement regarding the safety and effectiveness of sympathetic chain clamping is based not only on HRV data, but also on the post-operative satisfaction questionnaire results and the absence of systemic dysautonomic disorders, as reported in the abstract: “No development of systemic dysautonomic disorders was seen after the surgery.” (page 1, lines 29-30). These aspects are further detailed in the Results section of the manuscript. However, we understand the concern and have revised the last line of the abstract subsection Result to better reflect the nature of the findings.

Comment 5: Introduction: The investigation of the problem seems too old as references are limited up to 2023, most of them published before 2020. I strongly recommend update of the research analysis of published literature in order to show the state-of-the-art developments in recent years 2024, 2025, considering that the great interest to the automated HRV analysis in the era of AI.

Response 5: We thank the reviewer for the valuable suggestion. Following your advice, we performed an updated literature search focusing on publications from 2024 and 2025 regarding automated HRV analysis, particularly in the context of sympathetic chain clamping or related surgical procedures. Despite this effort, we were unable to identify any additional studies directly relevant to our specific topic beyond those already cited, the most recent being from 2023. However, to emphasize that our manuscript reflects a thorough and current review, we have added a statement in the Introduction noting that no relevant studies from 2024 or 2025 were found (page 2, line 64). If the reviewer is aware of any key publications from this period that we may have overlooked, we would appreciate your guidance in including them. Thank you.

Comment 6: Introduction: “The production and secretion of sweat are controlled by the parasympathetic arm of the autonomic nervous system (ANS).” -> Please revise the statement as there are evidences that the production and secretion of sweat are primarily controlled by the sympathetic branch of the autonomic nervous system, specifically via cholinergic fibers innervating the eccrine sweat glands. Although acetylcholine is typically associated with the parasympathetic system, in the case of sweating it is released by sympathetic postganglionic neurons, making this a notable exception.

Response 6: Thank you for pointing this out. We acknowledge that this issue was due to an oversight, and we have now corrected it in the revised version of the manuscript, as can be found on page 1 line 39.

Comment 7: Ln 60-64: The aims of the study do not consider observation of the effect of the thoracoscopic sympathetic chain clamping as there are evidences for that in the Abstract. Please, revise the aims to represent the different focuses of the study. One option is to distinguish between primary and secondary objectives. Additionally, the aims should highlight the expected clinical benefits or implications of the findings to better justify the relevance and significance of the research.

Response 7: We appreciate the reviewer’s insightful comment. The necessary correction has now been made in the revised manuscript, as can be seen on page 2 lines 67-69.

Comment 8: Ln 75: “high quality ECG signals” -> Please clarify what is meant by this inclusion criterion. The quality of an ECG primarily reflects the acquisition conditions rather than the patient’s clinical status. Since high-quality ECGs can generally be obtained at any time, this criterion requires further explanation. Additionally, the specifics of ECG acquisition—such as the equipment used and recording conditions—should be described before applying this criterion.

Response 8: We agree that the expression "high quality ECG signals" was not sufficiently specific. In the revised manuscript, we have clarified that. This change has been made in the [Methods section, page 2, line 85-87].

Comment 9: 2.1. Study Design” -> Provide information about the center involved in this single-center trial.

Response 9: Thank you for pointing this out. We added this information in the text, on page 2 line 72.

Comment 10: Section “2.1. Study design”: provide information about Ethical approval of this study (Institution, ID and date) and its registration in an official register.

Response 10: We thank the reviewer for highlighting this important aspect. We provided this information on page 2 lines 77-80.

Comment 11: Section “2.2. Patients”: The enrollment process and testing procedures for healthy controls are not described. Since this group serves as the reference, detailed information about their selection and assessment should be clearly provided.

Response 11: We thank the reviewer for this valuable comment. In the revised manuscript, we have clarified the selection process for the healthy control group, including inclusion and exclusion criteria, as well as the ECG quality standards applied. The same methodological rigor used for the patient group was also applied to controls, ensuring comparability. This clarification has been added to the Methods section (page 3, lines 91–93).

Comment 12: Section “2.3.1. ECG and HRV Recording” -> Please provide detailed information about all devices and software used, including their version numbers and the country of origin of the manufacturers.

Response 12: We thank the reviewer for the comment. The requested information is already included in the manuscript (page 3, lines 98–102). We hope this clarifies the issue.

Comment 13: Ln 84-85: “KardiaMobile® 6L (AliveCor Inc.) device with a five minute recording in a quiet room.” -> Please explain why recording in a quiet room is important and whether it relates specifically to the transmission of ECG data as audio frames. Otherwise, the absence of ambient noise is not a standard requirement for ECG acquisition, which typically emphasizes a resting state rather than environmental sound conditions.

Response 13: We thank the reviewer for this comment. The mention of a "quiet room" was not related to the technical transmission of ECG data as audio frames, but rather to the physiological context of the recording. We aimed to minimize external stimuli, including noise, as they may influence the autonomic nervous system and thus affect heart rate variability (HRV) measurements. Ensuring a calm and quiet environment was therefore intended to help maintain a true resting state and improve the reliability of HRV data. To clarify this point, we have revised the sentence in the manuscript accordingly, as can be seen on page 3 lines 99-102.

Comment 14: Ln 87-88: “Each recording was double-checked singularly.” -> Please, explain the statement.

Response 14: We agree that the original sentence was unclear. We have now rephrased it to clarify that, after automatic preprocessing, each HRV recording was manually reviewed by an operator to exclude possible residual errors. The revised sentence is on page 3, lines 104-105.

Comment 15: Ln 89: All HRV features must be explained with their full names instead of directly using abbreviations.

Response 15: Thank you for your advice. We provide a better explanation of these features by expressing their full name (page 3 lines 107-109).

Comment 16: Ln 91-93: “Parasympathetic nervous system (PSN) and Sympathetic nervous system (SNS) indexes were obtained from integrated computations.” -> The statement requires further clarification. Please provide more detailed information on how these indexes were computed, as they are not standard measures. Including references, formulas and computational methods would help readers better understand the evaluation process.

Response 16: We appreciate the reviewer’s comment. We provided a better clarification to specify how PSN and SNS indexes were computed (page 3, line 111-115).

Comment 17: In Tables 2, 3, 5 many HRV features are reported, however they are NOT defined in Methods. This lacks interpretation of these features in results. Typically, all HRV features should be clearly defined in Methods, ideally accompanied by an example of the HRV analysis process—such as software used, illustrative plots (Time-domain and frequency-domain), and details of spectral computations - to clarify how these features are derived. Currently, the HRV analysis appears as a black box, making the reported features difficult to interpret.

Response 17: We thank the reviewer for this valuable comment. We agree that defining each HRV feature in detail is important for interpretability. However, to maintain conciseness and avoid excessive redundancy, we opted not to include a full description of each individual parameter in the Methods section. Instead, we added a general statement referring to the well-established nature of HRV analysis and cited key literature where these features, their physiological meaning, and their applications are extensively described (page 1 line 59-60). However, we have also clarified this point in the revised text to better guide the reader, by adding a summary Table in the Appendix of our paper.

Comment 18: Ln 93: “ECG features were also sampled.” -> The statement requires further clarification, including the computed ECG features under interest.

Response 18: Thank you for pointing that out. We expressed more clearly in the revised version of the manuscript, as you can see on page 3, lines 118-119.

Comment 19: Ln 95-96: “Selective Clamping of the Thoracic Nerve in Video-Thoracoscopy” -> Provide reference(s) that define the standard protocols or guidelines for this procure to support its validity.

Response 19: We appreciate your suggestion. Indeed, we provided a reference of the standard surgical protocol, upon which we based our procedure.

Comment 20: Section “2.4. Outcomes” -> This section should be expanded to include clear and precise definitions of the primary and secondary outcomes. The term “aim” is inappropriate here, as the study aims are already defined elsewhere, so avoid redundant use. It is also unclear why only the “ANS evaluation of PFHH patients” is mentioned, given that healthy controls serve as a reference group. Please clearly define the groups involved in the outcome analyses, including comparisons between PFHH patients and healthy controls, as well as pre- versus post-operative PFHH patients, specifying the features under investigation for each groups.

Response 20: We thank the Revisor for the helpful comment. As suggested, we have rephrased the paragraph to clearly define the primary and secondary outcomes, specifying the comparisons between PFHH patients and healthy controls.

Comment 21: Section “2.5 Statistical Analysis” -> Provide power analysis to plan the number of required patients to validate the primary hypothesis.

Response 21: We thank the Reviewer for this important observation. Therefore, we have added a priori power analysis to Section 2.5 “Statistical Analysis” (lines 168-173), based on the expected effect size of the primary outcome.

Comment 22: Ln 125-126: “For a small group of patients, pre-second intervention HRV data were available, and so they were compared with pre-first intervention ones. “ -> This statement is confusing as pre-first and pre-second interventions must be defined before in methods. Clarification is strongly required.

Response 22: We thank the Reviewer for pointing this out. The section mentioned was originally part of an earlier draft of the manuscript and was intended to be removed during revision. Unfortunately, it was inadvertently left in the submitted version. We have now corrected this and removed the passage in the revised manuscript.

Comment 23: Results: Please consider including a CONSORT-style flow diagram to clearly illustrate the progression of patients from enrollment through to final analysis in groups (do not forget pre-first and pre-second interventions), taking into account all inclusion and exclusion criteria. This would enhance transparency regarding patient selection and clarify how the final study population was determined.

Response 23: We appreciate your suggestion. We clarify the enrollment process in the revised version of the manuscript.

Comment 24: Ln 137: The term “PFHH patients (case group)” is unclear. It is preferable to use a consistent and meaningful term such as “PFHH group” throughout the text. Additionally, specify “pre-operative PFHH group” when appropriate to clearly distinguish it from the “post-operative PFHH group” later in the manuscript.

Response 24: Thank you for clarifying that. We fixed that issue.

Comment 25: Results: Please include the sample size (number of participants) for each group in all tables, as this information is essential for proper interpretation of the results.

Response 25: Again, thank you for your suggestion. We fixed that issue.

Comment 26: Table 2, Table 3, Table 5 and corresponding text: Limit all numerical values to three or four significant digits - for example, round 1098.86 to 1099, or 47.09 to 47.1. This will enhance readability and eliminate unnecessary precision that does not add meaningful information.

Response 26: We agree with your comment. Indeed, we have revised the Tables accordingly.

Comment 27: The results in Table 2 are very interesting. The data confirm that patients with PFHH show reduced sympathetic modulation (as evidenced by significantly lower LF and VLF power), while parasympathetic activity remains unchanged. There is no evidence of parasympathetic dominance, but rather of reduced autonomic response overall, especially on the sympathetic side. These results are consistent with the idea of ANS dysregulation but not of overactive resting sympathetic tone - contrary to what might be assumed based on the pathophysiology of PFHH. These data do not support the hypothesis of resting sympathetic hyperactivity in PFHH. The observed autonomic imbalance may reflect altered central regulation in PFHH, although localized sympathetic overactivity (e.g., to sweat glands) may still be present but not detectable through global HRV metrics.

Response 27: Thank you for your insightful and accurate interpretation. We agree with you opinion and we discussed this and other hypotesis in the Discussion section.

Comment 28: The results in Table 3 are also very interesting. The post-operative changes show that Heart rate ↑, Time-domain HRV ↓, PNS index ↓, SNS index ↑, and moderate (but non-significant) HF power ↓. These findings point to a significant shift toward sympathetic dominance and vagal withdrawal, likely reflecting the autonomic consequences of thoracoscopic sympathetic chain clamping. Importantly, while time-domain and autonomic indexes show strong and significant effects, frequency-domain measures do not reach significance, possibly due to limited sample size or measurement variability.

Response 28: Again, thank you for your insightful and accurate interpretation. We agree with you opinion and we discussed this and other hypotesis in the Discussion section.

Comment 29: Table 5, caption -> Not informative.

Response 29: thank you for pointing that out. We revised the caption and fixed that issue.

Comment 30: Results: As there is no restriction on the number of figures or tables, I recommend including graphical representations to enhance the visualization of key findings from the most important tables. This would improve data interpretation and make group comparisons clearer for the reader.

Comment 31: Section “Author Contributions” -> Must be rewritten according to the journal’s guidelines, described here: https://www.mdpi.com/journal/medsci/instructions, copy of the text: “For research articles with several authors, a short paragraph specifying their individual contributions must be provided. The following statements should be used "Conceptualization, X.X. and Y.Y.; Methodology, X.X.; Software, X.X.; Validation, X.X., Y.Y. and Z.Z.; Formal Analysis, X.X.; Investigation, X.X.; Resources, X.X.; Data Curation, X.X.; Writing – Original Draft Preparation, X.X.; Writing – Review & Editing, X.X.; Visualization, X.X.; Supervision, X.X.; Project Administration, X.X.; Funding Acquisition, Y.Y.””

Response 31: Thank you. We fixed that issue.

Reviewer 2 Report

Comments and Suggestions for Authors

This is a well-conducted observational case-control study that explores the autonomic cardiac function, assessed by heart rate variability (HRV), in patients with primary focal hyperhidrosis (PFHH) undergoing thoracoscopic sympathetic chain clamping. The study is notable for its relatively large sample size and its comprehensive assessment of both pre- and post-surgical autonomic profiles, including a follow-up survey on compensatory hyperhidrosis and dysautonomic symptoms. The manuscript is generally well written, and the clinical question is relevant and underexplored in current literature.

Major Comments:

  1. The interpretation of increased post-surgical SNS index and decreased PNS index should be more nuanced. The authors propose a compensatory mechanism preoperatively, but this remains speculative. Including a brief mechanistic discussion or citing experimental data supporting this interpretation would strengthen the discussion.
  2. While the control group was age-matched, more detail should be provided regarding how these participants were recruited and screened for autonomic or cardiovascular conditions. This is critical for validating the control group as a proper comparator.
  3. While some exclusion criteria were used (medications, or arrhythmias), it is unclear whether factors like caffeine, physical activity, or circadian variation—known to influence HRV—were controlled during the ECG acquisition. This should be clarified.
  4. The subjective nature of CH assessment via self-reported survey is acknowledged, but more details on the validation of this questionnaire or the specific scoring system used would improve the transparency of the outcome assessment.

Author Response

We are grateful to the reviewer for his/her thoughtful and detailed suggestions, which have significantly contributed to enhancing the manuscript.

Comment 1: The interpretation of increased post-surgical SNS index and decreased PNS index should be more nuanced. The authors propose a compensatory mechanism preoperatively, but this remains speculative. Including a brief mechanistic discussion or citing experimental data supporting this interpretation would strengthen the discussion.

Response 1: We appreciate the reviewer’s comment. We acknowledge that, for now, we are unable to provide empirical data to directly support our hypothesis. As such, we have clearly presented it as a speculative interpretation, aimed at offering a possible explanation that could guide future investigations. We have also revised the manuscript to ensure this is explicitly stated, as highlighted on page 10 lines 275-278.

Comment 2: While the control group was age-matched, more detail should be provided regarding how these participants were recruited and screened for autonomic or cardiovascular conditions. This is critical for validating the control group as a proper comparator.

Response 2: We thank the reviewer for this valuable comment. In the revised manuscript, we have clarified the selection process for the healthy control group, including inclusion and exclusion criteria, as well as the ECG quality standards applied. The same methodological rigor used for the patient group was also applied to controls, ensuring comparability. This clarification has been added to the Methods section (page 3, lines 91–94).

Comment 3: While some exclusion criteria were used (medications, or arrhythmias), it is unclear whether factors like caffeine, physical activity, or circadian variation—known to influence HRV—were controlled during the ECG acquisition. This should be clarified.

Response 3: Thank you for your thoughtful observation. While we attempted to standardize recording conditions (e.g., using a quiet environment and consistent timing), we acknowledge that these specific factors were not strictly monitored in our study. We have now added a sentence to the manuscript to explicitly recognize this as a limitation and to suggest it as a consideration for future research, as stated on page 11 lines 317-321.

Comment 4: The subjective nature of CH assessment via self-reported survey is acknowledged, but more details on the validation of this questionnaire or the specific scoring system used would improve the transparency of the outcome assessment.

Response 4: Thank you for your helpful comment. To improve the transparency of the outcome evaluation, we have added the full post-surgical satisfaction questionnaire used in the study as Supplementary Material. We have also clarified this point in the revised manuscript.

Reviewer 3 Report

Comments and Suggestions for Authors

Dear authors of the medsci-3772037 paper, I have some constructive comments on your work.

Section 2.1 Methods: I suggest you indicate the "type of data" you collected.

Section 2.3.1 Methods: Here you describe how you obtained the "patient data." Why don't you indicate how you obtained the control subjects' data?

Line 162: Please describe the abbreviation CH.

Was the work approved by a local ethics committee?

Author Response

We are grateful to the reviewer for his/her thoughtful and detailed suggestions, which have significantly contributed to enhancing the manuscript.

Comment 1: Section 2.1 Methods: I suggest you indicate the "type of data" you collected.

Response 1: Thank you for your comment. We agree that the term "data" is broad; however, we clarify and detail the type and nature of the data in the following sections of the manuscript. To address your concern, we have slightly rephrased the sentence to indicate that a more detailed description follows.

Comment 2: Section 2.3.1 Methods: Here you describe how you obtained the "patient data." Why don't you indicate how you obtained the control subjects' data?

Response 2: Thank you for your valuable suggestion. We have now revised the text to explicitly indicate that the same acquisition protocol and quality criteria were applied to the control group, as highlighted on page 3 lines 91-94 and 101-102.

Comment 3: Line 162: Please describe the abbreviation CH.

Response 3: Thank you for pointing that out. We added the acronym near the full name on page 7 line 202, in order to be more explicative.

Comment 4: Was the work approved by a local ethics committee?

Response 4: Thank you for your accurate comment. Yes, our work was approved by the local ethics committee. We provided all information in the revised version of the manuscript (page 2, Lines 77-80).

Round 2

Reviewer 1 Report

Comments and Suggestions for Authors

The authors have correcrtly communicated all my revision comments and questions. I have no further remarks. The methods and findings presented in the study offer valuable contributions and support its suitability for publication.